# A World without Measles and Rubella: Addressing the Challenge of Vaccine Hesitancy

**DOI:** 10.3390/vaccines12060694

**Published:** 2024-06-20

**Authors:** David M. Higgins, Sean T. O’Leary

**Affiliations:** Adult and Child Center for Outcomes Research and Delivery Science (ACCORDS), Children’s Hospital Colorado, University of Colorado School of Medicine, Aurora, CO 80045, USA; david.higgins@cuanschutz.edu

**Keywords:** vaccination, vaccine hesitancy, vaccine delay, vaccine refusal, measles, rubella

## Abstract

The worldwide elimination of measles and rubella is feasible, but not without overcoming the substantial challenge of vaccine hesitancy. This challenge is complicated by the spread of misinformation and disinformation fueled by rapidly progressing technologies and evolving forms of online communication. The recent COVID-19 pandemic has only added further complexity to this challenge. However, considerable progress has been made in understanding the scope of the problem and the complex factors that influence vaccine hesitancy. Our understanding of evidence-based strategies for addressing vaccine hesitancy has grown significantly, including evidence for effective communication and behavioral interventions. In this article, we review measles and rubella vaccines and vaccine hesitancy. We then provide an overview of evidence-based strategies for addressing vaccine hesitancy, including communication strategies and behavioral interventions. This article is relevant to healthcare professionals, health system leaders, public health professionals, policymakers, community leaders, and any individuals who have a role in addressing vaccine hesitancy in their communities. Finally, we review future directions and major areas of research need.

## 1. Introduction

The measles and rubella viruses continue to cause over 9 million cases of disease and 136,000 deaths worldwide every year despite the availability of effective, safe, and inexpensive vaccines for over half a century [1]. As obligate human pathogens, eradicating measles and rubella worldwide is technically feasible. However, despite all six World Health Organization regions committing to elimination, no region has achieved and maintained elimination. Disruptions to vaccination programs during the coronavirus-19 (COVID-19) pandemic resulted in significant setbacks to measles and rubella elimination efforts, which have yet to be reversed [1].

Efforts to eradicate measles and rubella from the world must address the substantial challenge of vaccine hesitancy to be successful. Vaccine hesitancy, defined by the WHO as a “delay in acceptance or refusal of vaccines despite availability of vaccination services”, can lead to measles and rubella vaccine delay or refusal, which undermines elimination efforts [2,3]. Factors influencing vaccine hesitancy are complex and involve historical, sociocultural, environmental, institutional, economic, political, and individual/group factors [2]. These factors are further influenced by rapidly changing technologies that allow for the rapid and ubiquitous spread of information, misinformation, and disinformation. Fortunately, since vaccine hesitancy has been around since the development of vaccines, there is a long history of experience and evidence to inform efforts to address vaccine hesitancy.

In this article, we review the history and current state of measles and rubella vaccine hesitancy. We then provide an overview of evidence-based strategies for addressing vaccine hesitancy, including communication strategies and behavioral interventions that can be implemented within systems and organizations. These strategies are relevant to healthcare providers, health system leaders, public health professionals, policymakers, community leaders, and any individuals who have a role in addressing vaccine hesitancy in their communities. Finally, we review future directions and major areas of need for research.

With effective, safe, and inexpensive vaccines, every illness, complication, or death from measles and rubella is unacceptable. Using evidence-based strategies to address vaccine hesitancy and a re-commitment worldwide to improve vaccine uptake, measles and rubella can be eradicated.

## 2. History of Measles and Rubella Disease and Vaccination

Before the development and widespread availability of vaccines in the late 20th century, measles and rubella plagued humanity for thousands of years, with no effective tools to prevent their consequences [4,5]. Measles disease was first described in the 7th century, although the infectious pathogen would not be identified until the early 20th century [5]. Measles is a highly contagious viral disease that classically begins with a prodrome of fever, cough, coryza, conjunctivitis, and spots on the buccal mucosa (Koplik spots), followed by a maculopapular rash which starts on the head and neck and spreads to the rest of the body [6]. While some individuals have a resolution of these symptoms, many will develop complications, including diarrhea, otitis media, pneumonia, encephalitis, subacute sclerosing panencephalitis, or death. Rubella was initially thought to be a variant of measles but was first identified as a separate disease in 1814 [4]. Rubella is a viral disease that begins with a prodrome of mild fever, malaise, lymphadenopathy, and upper respiratory symptoms before the appearance of a maculopapular rash [6]. While complications of rubella are rare in many individuals, for pregnant persons, rubella infection can lead to congenital rubella syndrome (CRS), which can result in miscarriages, stillbirths, and severe birth defects in infants [6].

For centuries, measles and rubella have caused significant public health burdens worldwide. Measles is estimated to have caused greater than 2–3 million deaths worldwide every year before the availability of vaccines [5]. While precise historical estimates of the public health burden of rubella are lacking, during the last major rubella epidemic in the United States in 1964–1965, there were an estimated 12.5 million rubella cases with 30,000 affected pregnancies leading to 6250 spontaneous abortions and 2100 newborns who died at or shortly after birth [4]. Another 20,000 children were born with CRS, resulting in approximately 11,600 infants who were deaf, 3580 who were blind, and 1800 who were mentally disabled.

Scientific advancements in the 18th, 19th, and 20th centuries culminated in the successful development of safe and effective vaccines against both measles and rubella [4,5]. Two measles vaccines (an inactivated and live attenuated vaccine) were developed and licensed in the United States in 1963 [5]. The inactivated vaccine was later withdrawn due to poor effectiveness. Over the next five years, further attenuated live measles vaccines were developed and licensed, showing improved side-effect profiles over the originally licensed live attenuated vaccine [5]. Four rubella vaccines were developed and licensed in the United States and Europe during the 1960s–1970s [4]. Eventually, one rubella vaccine became the predominant vaccine licensed and available worldwide due to superior effectiveness and a preferable side-effect profile. In 1971, the measles and rubella vaccines were licensed as a combined measles, mumps, and rubella (MMR) vaccine [5]. Later, in 2005, the combined measles, mumps, rubella, and varicella (MMRV) vaccine was licensed [5].

Aside from clean water, vaccines arguably reduced more diseases and deaths worldwide than any other public health achievement in human history, and measles and rubella vaccines are important contributors to this accomplishment [7]. With the widespread availability of measles and rubella vaccines and improvements in worldwide vaccination rates in the early 21st century, significant progress has been made toward eliminating measles and rubella [1]. From 2000 to 2022, the measles vaccine alone is estimated to have prevented 57 million deaths worldwide [1]. By 2022, 51% of the world’s countries had eliminated rubella disease [8].

Despite the incredible success of measles and rubella vaccination programs worldwide, these diseases remain a significant public health threat. It is estimated that measles caused 136,200 deaths worldwide in 2022, and rubella caused an estimated 32,000 cases of CRS in 2019 [1,9]. Gains in worldwide measles and rubella vaccination experienced significant setbacks during the COVID-19 pandemic, which have yet to recover [1,8]. The setbacks experienced during the COVID-19 pandemic are due to various factors, including major disruptions in measles and rubella vaccination programs and changes in attitudinal barriers to vaccination, including vaccine hesitancy. With widely available, effective, low-cost vaccines, the eradication of measles and rubella is technically feasible, but first, the challenge of vaccine hesitancy must be considered and addressed.

## 3. History and Current State of Measles and Rubella Vaccine Hesitancy

### 3.1. Brief History of Measles and Rubella Vaccine Misinformation and Antivaccine Activism

Vaccine hesitancy is primarily fueled by antivaccine activism, which has a history as long as vaccines themselves (Figure 1). Shortly after the introduction and use of the smallpox vaccine in the late 18th and early 19th centuries, opposition to vaccination was first led, in part, by medical professionals on the fringes of the medical and health professions [2]. This opposition to smallpox vaccination led to dropping vaccination rates, the resurgence of smallpox outbreaks, and the establishment of compulsory vaccination laws in the 19th century [2]. These laws differed from modern mandatory vaccination laws, which impose penalties for not vaccinating, as compulsory vaccination meant individuals had no choice in whether they received a vaccine. In response to enacting these compulsory vaccine laws, the antivaccine movement capitalized on perceived violations of personal freedoms, a tactic still used today, and successfully repealed some compulsory smallpox vaccination laws [2].

The middle of the 20th century saw a relative lull in the antivaccine movement, with improvements in the manufacture and delivery of vaccines and public health infrastructure [2]. However, in the 1970s and 1980s, the antivaccine movement again gained traction with the spread of both misinformation and disinformation about whole-cell pertussis vaccines and the establishment of several antivaccine organizations by antivaccine activists in the U.S., Australia, and Europe [2]. Misinformation is defined as false information that is shared by people who do not realize it is false and do not mean any harm, whereas disinformation is false information that is spread with malicious intent [10].

Within the context of this revitalized antivaccine movement came arguably the most damaging modern-day antivaccine misinformation incident. In 1998, Andrew Wakefield, a since-discredited British physician, published an article in the Lancet suggesting a link between the MMR vaccine and autism in 12 children [11]. This article was later found to be false and fraudulent [12]. The journal eventually retracted this now-infamous article. Andrew Wakefield was stripped of his license to practice medicine, and many large studies have confirmed there is no causal link between MMR and autism; the damage from this fraudulent article was already done [12]. The article received widespread media attention, and antivaccine activists jumped at the opportunity to add this misinformation to their reasons for vaccine opposition [13]. The fallout from this incident resulted in significant declines in MMR vaccination and inestimable preventable harm around the world.

Complicating the MMR and autism myth during the 1990s were concerns about the safety of thimerosal, a mercury-based preservative used in some vaccines [13]. Although there was no evidence that the dose of thimerosal in vaccines causes significant harm, in 1999, the U.S. Public Health Service agencies, the American Academy of Pediatrics, and vaccine manufacturers agreed to reduce or eliminate the use of thimerosal in vaccines [14,15]. While this action in the U.S. was taken as a precautionary measure, it sent mixed messages to the public about the safety of vaccines and added to antivaccine controversies. In the U.S., thimerosal was removed from all vaccines by 2001, except multi-dose influenza vaccine vials [14]. It is still used throughout the world with no evidence of harm.

Throughout the start of the 21st century, the spread of vaccine misinformation, disinformation, and antivaccine activism has been fueled by rapidly progressing technologies and evolving forms of online communication. In 2002, researchers studied online antivaccination information and concluded, “There is a high probability that parents will encounter elaborate antivaccination material on the world wide web. Factual refutational strategies alone are unlikely to counter the highly rhetorical appeals that shape these sites” [16]. Since then, with advancements in online technology and the explosion of social media use, vaccine misinformation and disinformation have spread rapidly. As Larson et al. noted in an editorial in 2022, “The role of social media in fueling the spread of vaccine hesitancy and its increasingly documented health consequences cannot be overstated” [17].

Although antivaccine sentiments and vaccine hesitancy have been studied and reported on most frequently in the US, similar patterns have taken shape worldwide, including in low- and middle-income countries (LMICs) [2,18]. Using data from 290 surveys conducted between 2015 and 2019 across 149 countries, researchers found varying determinants of vaccine hesitancy between countries, including perceptions of the importance, safety, and effectiveness of vaccines and changes in vaccine confidence over time [18]. Ultimately, the rising global threat of antivaccine sentiments and vaccine misinformation contributed to the WHO classifying vaccine hesitancy as a top ten global health threat in 2019 [19].

**Figure 1 vaccines-12-00694-f001:**
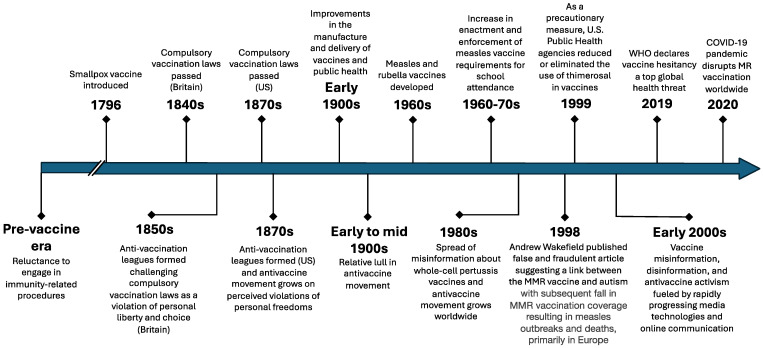
Timeline of significant events in vaccine hesitancy [2,20].

### 3.2. Definitions of Vaccine Hesitancy, Confidence, Acceptance, and Refusal

The clarity of terms when considering vaccine hesitancy is critical. Mixing terms or unclear definitions may add to confusion about vaccine attitudes and intentions and undermine attempts to communicate productively on these topics, especially when terms are used in public discourse [21,22]. There must also be an appreciation that terms such as vaccine hesitancy occur on a spectrum of attitudes and intentions, which is challenging to define. Attempts to have productive discussions about vaccine hesitancy should clarify terminology before use and recognize the common and practical uses and public perception of terms.

While many definitions for vaccine hesitancy have been proposed or reported in the literature, the WHO Strategic Advisory Group of Experts (SAGE) working group on vaccine hesitancy defined vaccine hesitancy as “…delay in acceptance or refusal of vaccination despite the availability of vaccination services. Vaccine hesitancy is complex and context-specific, varying across time, place, and vaccines. It is influenced by factors such as complacency, convenience, and confidence [3]”. As Bedford et al. point out, due to its wide use in public discourse, the term vaccine hesitancy may incorrectly be assumed to be a behavior when it is a psychological state that influences behavior or inaccurately used to explain all undervaccination in a population when other causes may be responsible, such as access barriers [22].

According to the WHO SAGE working group on vaccine hesitancy, vaccine confidence refers to vaccine safety and efficacy, the healthcare workers delivering vaccines, and those making decisions regarding vaccine approval in a population [2,3]. Other definitions of vaccine confidence include confidence in the processes and policies that lead to vaccine development, licensure, manufacturing, and use recommendations [23]. Vaccine acceptance has been described as “the degree to which individuals accept, question or refuse vaccination [24]”, and vaccine refusal as an “unwillingness to allow oneself or family member to be immunized against a preventable contagious disease [2]”. Other descriptions of vaccine acceptance and refusal have been proposed [2].

### 3.3. Common Determinants of Vaccine Hesitancy

The WHO categorized the determinants of vaccine hesitancy into a framework that includes three major categories: (1) contextual influences, (2) individual and group influences, and (3) issues related directly to a vaccine or vaccination [3]. The contextual influences include the media environment, leaders and anti- or pro-vaccine figures, historical perspectives, and religious, cultural, socio-economic, political, or geographic influences. Individual and group factors include personal, family, or community experiences with vaccination, general beliefs and attitudes about health and prevention, knowledge and awareness about vaccination, trust in health systems and professionals, perceived risks/benefits of vaccines, and social norms. Issues related directly to a vaccine or vaccination include the scientific evidence, introduction of new vaccines, the mode of administration, type of vaccination program, vaccine schedule, costs, and the knowledge, attitudes, and strength of recommendation of healthcare professionals.

These determinants of vaccine hesitancy can negatively or positively influence vaccine attitudes. Also, for each individual or community, multiple determinants work in complex ways to influence attitudes and intentions around vaccines. For example, media may be a source of widespread vaccine misinformation, but it also can be used to communicate important information regarding vaccination. Synergistically, either anti- or pro-vaccine leaders may use media or other contextual factors to enhance the influence of these determinants. Furthermore, the clustering of religious, cultural, and individual or group factors can create communities with unique vaccine hesitancy determinants influencing their attitudes and intentions. Therefore, any evaluation of the determinants of vaccine hesitancy in a population or individual should consider the broad range of factors that contribute to attitudes and intentions. Assuming a community is influenced by only one of these determinants alone may prevent the optimal use of strategies to address vaccine hesitancy.

### 3.4. Measurement of Vaccine Hesitancy

Vaccine hesitancy falls on a spectrum of vaccine attitudes and intentions. Several typologies of parental vaccine hesitancy have been described [2,25,26,27]. These vaccine hesitancy typologies generally span from those who recognize the importance of vaccines and accept all vaccines on one end of the spectrum to those who refuse all vaccines, often have mistrust in the medical system, or have other strong religious or personal beliefs, and their beliefs about vaccines are fixed [27]. These typologies help to describe general observations about the spectrum of vaccine hesitancy and can be used as a tool for tailoring vaccine communication; however, they do not entirely describe the heterogeneity found within the spectrum of vaccine hesitancy. Also, these typologies may evolve and must be adapted for different population contexts. Additionally, these typologies are not unique to measles and rubella vaccines. Furthermore, if these typologies are inappropriately used to mislabel individuals, this may inhibit honest and respectful communication.

Multiple vaccine hesitancy and confidence measures have been developed, and some of them validated, to describe where individuals are on the spectrum of vaccine hesitancy attitudes and intentions [2]. Some of the most commonly used and validated measures include the Parent Attitudes about Childhood Vaccines (PACV) [28], Vaccine Confidence Scale (VCS) [29], Vaccine Hesitancy Scale (VHS) [30], 5C Antecedents of Vaccine Acceptance (5C) [31], and Caregiver Vaccination Attitudes Scale (CVAS) [32]. It is important to note that these scales have limited validation studies except for the PACV and VCS [2]. Also, most of these measures were developed in high-income countries (HIC), and their utility in other populations has yet to be validated. Additionally, it is worth noting that these tools are not specific to measles or rubella vaccines. While an adaptation of the VHS survey tool for measuring measles vaccine confidence was used (aMVHS), it was limited in predicting a child’s measles vaccination status [33].

In general, these vaccine hesitancy measurement tools use at least several questions to measure some combination of constructs, including belief in the importance of vaccines, trust in vaccines and the professionals and processes that approve and recommend vaccines, perceived safety of vaccines, perceived risks of infectious diseases, and responsibility to the greater community [2]. Most of these measures’ conceptual basis is grounded in the Health Belief Model, where individuals’ vaccination behaviors are influenced by their perceptions of the vaccine and vaccine-preventable disease [34]. Unfortunately, most vaccine hesitancy measurement tools have limited psychometric validation and replication to ensure they measure what they are designed to measure [2]. Further studies validating these tools in different populations and correlating measurements with actual vaccination behavior are warranted, and caution should be used in interpreting these measurements.

Finally, the measurement of vaccine hesitancy at the population level is often reported by polling with methodologies that may or may not have been validated and are not always transparent [35,36,37]. These polls still have value in informing trends in vaccine hesitancy but are often misinterpreted by the media or the general public to reflect something they may not have measured. For instance, a poll asking a single question about support for vaccine requirements to attend school may be misinterpreted as reflecting the full spectrum of vaccine hesitancy attitudes in a population. Unfortunately, until validated vaccine hesitancy measures are more consistently applied and used, policymakers and health professionals are left with these data types to monitor vaccine hesitancy, guide policy, appropriately target populations with vaccine hesitancy interventions, and inform policy.

### 3.5. Epidemiology of Measles and Rubella Vaccine Hesitancy

A precise understanding of the epidemiology of measles and rubella vaccine hesitancy is inhibited by using different and sometimes unvalidated measurement tools, as described above, which are implemented inconsistently across populations and time. Additionally, vaccine hesitancy may vary by type of vaccination, and validated measures specific to measles and rubella vaccines are lacking and have not been widely used. Furthermore, since vaccine hesitancy is on a spectrum, classifying individuals as having vaccine hesitancy for epidemiologic assessment is challenging and varies between methodologies and measures. For these reasons, many countries or jurisdictions rely on vaccine uptake alone to estimate vaccine hesitancy in populations. However, these data cannot differentiate between attitudinal, behavioral, and access causes for undervaccination [2]. In response to these shortcomings, experts have called for standardized, timely, and actionable surveillance systems for vaccine hesitancy [38,39,40].

Despite these limitations, some epidemiologic data about the prevalence of vaccine hesitancy are critical to understand. U.S. studies have consistently shown that the proportion of parents with vaccine hesitancy leading to the outright refusal of all vaccines for their children is small (1–3%) [26,27,41,42]. These data suggest that the proportion of individuals with absolute fixed beliefs is small, and a majority fall somewhere on the vaccine hesitancy spectrum where they have potentially movable vaccine attitudes and intentions.

Based on a recent meta-analysis with studies from over 30 different countries, the worldwide cumulative prevalence of vaccine hesitancy among parents of children aged 0–6 years old has been estimated to be ~20% [43]. However, this prevalence varies greatly by country. Additionally, in this meta-analysis, there were variations in the definitions of vaccine hesitancy, a range of different measures used, differences in country income levels, and dissimilarities in the approved and recommended vaccines, leading to a high degree of heterogeneity among the studies. These global data mirror studies in the U.S., which estimate that the prevalence of overall vaccine hesitancy is around 20% [40,44].

There are ongoing global efforts to describe measles and rubella vaccine hesitancy and confidence at national levels [39,45]. However, for a disease as contagious as measles, even local pockets of vaccine hesitancy leading to undervaccination can result in disease outbreaks, and the evidence-based approaches to addressing vaccine hesitancy discussed below are difficult to implement at a national level. Therefore, more granular geospatial surveillance of vaccine hesitancy is necessary to have actionable data on vaccine hesitancy.

As stated previously, worldwide vaccination with measles and rubella vaccines decreased in the years following the start of the COVID-19 pandemic; however, it is unclear if this decrease is due primarily to disruptions in vaccination programs or significant changes in hesitancy toward measles and rubella vaccines. Several studies and polls in the U.S. and Canada have not shown significant changes in childhood vaccine hesitancy since the start of the COVID-19 pandemic [36,40,46,47]. On the other hand, global survey data demonstrated a decrease in the perception of the importance of vaccines for children in many countries before and after the start of the pandemic [48,49]. Further data are needed to fully understand the relationship between post-pandemic decreases in measles and rubella vaccination and changes in vaccine hesitancy. It is also critical to determine whether pandemic-related changes to vaccine hesitancy to measles and rubella vaccines, if present, are transient or long-lasting.

## 4. Communication Strategies to Improve Measles and Rubella Vaccine Acceptance

In light of the challenge of vaccine hesitancy described above, what can be done to overcome vaccine hesitancy and, ultimately, improve measles and rubella vaccine uptake? Over the past several decades, many approaches for individual and public communication and healthcare system, organizational, and public health strategies to address vaccine hesitancy have been recommended or implemented. However, rigorous evidence supporting these strategies was largely lacking until recently. There are multiple ongoing trials and studies to evaluate strategies to improve vaccine uptake, and our understanding is continually evolving and improving.

Despite the rising threat of vaccine hesitancy in LMICs, most of the research on communication strategies to address vaccine hesitancy has been conducted in HICs. Behavioral research on improving vaccine uptake in LMICs has primarily focused on promoting the demand for vaccinations through social mobilization which includes the involvement of a wide range of national and local partners to raise the demand for vaccines [50,51]. Studies from LMICs are included, where appropriate, to the strategies discussed below; however, there is a significant evidence gap for high-quality research on communication and behavioral strategies to address vaccine hesitancy in LMICs. Multiple studies in LMICs are ongoing that will improve the evidence for strategies in LMICs [51].

While more work to gather evidence for communication strategies is ongoing and needs to be completed, several strategies have more evidence than others in improving vaccine acceptance [2]. The following is not a comprehensive review of every strategy used but an overview of strategies with the most robust evidence that are most commonly used. Although all the strategies discussed here have not been studied for effectiveness in improving measles or rubella vaccine uptake specifically, the principles are likely generalizable to all childhood vaccines. Also, many of these strategies have been used in multi-component interventions, making it difficult to determine the isolated effectiveness of individual strategies.

### 4.1. Healthcare Professional Communication Strategies

As trusted sources about vaccines, healthcare professionals have the potential to influence vaccination attitudes and behaviors [52]. A recommendation from a healthcare professional is one of the most important predictors of vaccine acceptance [27].

Healthcare professionals should be prepared to answer vaccine-related questions or concerns and be knowledgeable about the facts related to vaccines and vaccine-preventable diseases. In populations where a lack of awareness or understanding of vaccination is a barrier, providing education and information about vaccines can improve uptake [53]. For instance, in some LMICs, studies have shown education and information about vaccines can improve vaccine uptake [51,54,55,56,57]. However, a growing body of evidence has shown that unless there is a significant knowledge gap, simply educating parents on the facts about vaccines may improve parents’ knowledge about vaccines but results in little to no difference in parental attitudes toward vaccines or intention to vaccinate [53,58].

Training healthcare professionals on vaccine communication strategies is critical, and many of these professionals report a need for further education [27]. However, training healthcare professionals on information about vaccines alone has not been found to improve vaccine uptake [59].

Establishing an honest dialogue, actively listening, and welcoming questions are all critical to the vaccination discussion [27]. Although many strategies have been proposed for healthcare professionals to communicate with vaccine-hesitant parents or patients, only in recent years have some strategies been rigorously tested in large, randomized controlled trials (RCTs) [27]. These strategies can be used concurrently or as needed based on the flow of the vaccine discussion. Additionally, although most frequently studied in clinicians in ambulatory healthcare settings, these strategies should translate to other health professionals who play a role in vaccination delivery processes in various settings. The following strategies have evidence supporting their effectiveness in improving vaccine uptake (Table 1) [27]:Providing a strong and high-quality recommendation;Using the presumptive format for initiating vaccine communication (“Johnny’s due for three shots today”);Pursuing adherence despite initial parental reluctance;Using motivational interviewing for parents or patients who express hesitancy.

There is good evidence that a strong recommendation for vaccination from a healthcare professional increases vaccine acceptance [60]. When a parent or patient hears a recommendation for vaccination, they are more likely to receive a vaccine than if no recommendation is made [61]. While this strategy may seem self-evident, studies have found that missed opportunities to recommend vaccination are frequent and correlate with decreased vaccine uptake [62,63,64]. The strength of the vaccine recommendation is also important and correlates with improved vaccine uptake [27].

Another communication strategy that can be used with a strong recommendation for vaccination is the presumptive approach when initiating vaccine communication, where vaccine communication is initiated with a closed-ended statement such as “Jack is due for several vaccines today” [65,66,67,68,69]. The presumptive approach contrasts with a participatory approach, where an open-ended question is used. Presenting vaccination as the default option presents vaccination as an opt-out decision, making it more likely for the parent or patient to stick to the status quo and accept vaccination [70,71]. Multiple studies have shown that the presumptive approach for initiating vaccination communication is associated with increased vaccine uptake, even among parents who have negative attitudes toward vaccines. Furthermore, clinicians report high satisfaction with using the presumptive approach and improvements in the efficiency of vaccine discussions [72]. Strongly recommending a course of action or using opt-out default language when communicating about any health intervention must only be performed when there is a clear low risk and high benefit to an intervention, such as with measles and rubella vaccination [73].

Another complementary strategy with some evidence supporting its effectiveness is pursuing adherence despite initial parental reluctance [27]. An example of this strategy would be to reemphasize the importance of a vaccine despite the parent expressing some reluctance to vaccinate. Additionally, despite non-acceptance at an initial visit, pursuing adherence with follow-up communication or repeating recommendations at future visits is associated with improved vaccine acceptance [74,75]

When individuals still express reluctance to vaccinate following a strong presumptive recommendation, motivational interviewing (MI) is another strategy that can be used [27]. Motivational interviewing is a patient-centered approach to enhancing behavior change by leveraging a patient’s inherent motivations [76,77]. The evidence for using MI when addressing vaccine-hesitant patients or parents is growing, and additional studies are ongoing [78,79,80,81,82,83]. Multiple RCTs have found that training providers on MI strategies improved HPV vaccine uptake [78,80,84].

Other proposed strategies exist with limited evidence to date including (1) health professionals using their own experiences with vaccine-preventable diseases or vaccines, (2) mentioning strategies available to minimize the pain associated with vaccination, and (3) bundling all vaccines that a child is eligible for at a visit with a single recommendation [27]. Additionally, including non-clinician personnel such as front desk staff, medical assistants, nurses, and other health professionals in all the vaccine communication processes discussed here may enhance their effectiveness and create a culture of vaccination in the healthcare setting. In LMICs, several studies have demonstrated that trained community members can use vaccine communication strategies to improve vaccine acceptance [54,56,85].

Finally, healthcare professionals should be prepared to answer vaccine-related misinformation or myths. There is emerging evidence to guide providers in carefully debunking a myth or misinformation that follows the fact, warning, fallacy, fact approach [27,86]. In this approach, debunking should start with the fact, followed by a warning about the myth, then briefly explaining the fallacy of the myth, followed by repeating the fact. It is also helpful to highlight the potential hidden motives of people who spread disinformation.

### 4.2. Individual Communication Strategies

With improvements in technology, there has been significant interest in individual or tailored communication strategies such as apps and web-based interventions to improve vaccine uptake with mixed success [2,87,88]. An RCT of delivering web-based vaccine messages tailored to parents’ vaccine attitudes did not improve infant vaccine uptake [89]. Multiple similar apps or websites have been developed; however, most of these did not evaluate or demonstrate improvements in vaccine uptake [90,91,92,93,94,95,96]. However, many of these apps or web-based tools have been shown to improve vaccine knowledge, attitudes, or beliefs. For example, a parent-centered, gamified mobile intervention about the MMR vaccine increased parents’ knowledge, intention to vaccinate, and confidence in the vaccination decision [94].

Further work is warranted on individual-level strategies using apps and web-based tools. Concerns about these tools’ usability, scalability, and availability for different populations and settings still need to be addressed. In addition to demonstrating positive effects on vaccine uptake, these dissemination, equity, and sustainability issues must be addressed.

### 4.3. Mass Communication Strategies

While there is broad interest in using mass communication (social media, website, TV, etc.) to improve vaccine attitudes and uptake, evidence for the effectiveness of messages and mass communication strategies is lacking [2]. Few large-scale and robust studies exist to inform whether these platforms actually improve vaccine uptake [88,97]. Unfortunately, some well-meaning strategies have even been shown to backfire among the most hesitant parents. For instance, a web-based intervention aimed to improve MMR vaccination by sharing one of four different types of information with parents (information explaining the lack of evidence that MMR causes autism; information about the dangers of measles, mumps, and rubella; images of children who have measles, mumps, and rubella; or a dramatic narrative about an infant who almost died of measles) increased misperceptions about the MMR vaccine or reduced vaccination intention [98].

Despite a lack of robust evidence, different mass communication strategies have been recommended, including the promotion of vaccines from healthcare professionals or other trusted messengers, the use of influencers or celebrities, using narratives, specifically targeting parents, and framing messages in a way that optimizes vaccination behavior change [99]. Unfortunately, due to anti-physician and anti-establishment sentiments and the algorithms that curate media echo chambers online, many of these strategies may not effectively reach the most vaccine-hesitant individuals or influence their attitudes or behaviors [99].

Unfortunately, the use of mass communication for spreading vaccine misinformation and disinformation, which contributes to vaccine hesitancy, is well documented [2,100]. The need for evidence-based strategies to address the spread of mis/disinformation through mass communication has been identified as a public health priority by the Surgeon General of the United States and WHO and is a critical area of future research [100,101,102].

## 5. Healthcare System, Organization, and Public Health Strategies to Improve Measles and Rubella Vaccine Acceptance

Beyond communication strategies to improve vaccine acceptance, multiple healthcare system, organizational, and public health strategies improve vaccine uptake [2]. These strategies may not be intended to address vaccine hesitancy directly; however, they may effectively overcome low to moderate levels of vaccine hesitancy by lowering or incentivizing the activation energy required for vaccination.

### 5.1. Healthcare System/Organizational Strategies to Improve Vaccine Uptake Reminder

There are a variety of healthcare systems or organizational strategies to improve vaccine uptake, and they are often used concurrently with communication interventions. For instance, these strategies decrease missed opportunities to make a strong presumptive recommendation for vaccination or increase the repeated use of those communication strategies. The following healthcare system/organizational strategies have evidence supporting their effectiveness in improving vaccine uptake [87]:Reminder/recall;Standing orders;Provider assessment/feedback;Provider reminders.

Reminder and recall systems encompass various methods to identify and remind or notify individuals or parents when vaccines are due. Reminder and recall can be conducted using mail, phone, text, apps, other media, and electronic health records. These notifications are sometimes tailored to individuals and can be accompanied by educational messages [92]. Both observational studies and RCTs in HICs and LMICs have shown evidence that these methods increase vaccine uptake and they are often combined with other strategies [51,103,104,105].

Standing orders allow healthcare professionals to administer vaccines according to a protocol approved by a supervising authorized practitioner. While no RCTs exist for this strategy, observational studies have demonstrated improved vaccine uptake [88]. Standing orders can reduce missed opportunities for vaccination and empower non-clinician healthcare personnel to have a significant role in vaccine delivery [88]. Most of the evidence for standing orders comes from the U.S.; contextual differences in healthcare infrastructure and authority for different healthcare professionals to administer vaccinations may limit its generalizability to other countries.

Provider assessment and feedback include strategies whereby healthcare professionals receive their vaccination rates and feedback on improvement [87,106]. Although several studies failed to demonstrate improved vaccine uptake with audit and feedback alone, other studies showed improvements in vaccine uptake, especially when used concurrently with other strategies [87,88].

Provider reminders include a variety of methods to remind providers about when patients are due for vaccinations [87,88]. Provider reminders can be performed in various ways, including written notes, chart flags, or electronic health record alerts. Multiple studies have shown the effectiveness of provider reminders for vaccine uptake, and this strategy is often combined with others [107,108,109].

Finally, immunization information systems (IISs), which serve as a confidential central source of vaccination information for a geographic area, can enhance the availability and use of multiple strategies, including provider reminders, audit and feedback, and reminder and recall strategies [88].

### 5.2. Strategies to Improve Access to Vaccination Services

While a comprehensive review of strategies to improve access to vaccination services is beyond the scope of this article, it is important to recognize that the use of the evidence-based strategies discussed above depends on access to high-quality vaccination services such as health professionals with adequate training on vaccine communication and healthcare systems or organizations that can use evidence-based methods to promote vaccination. Individuals are unlikely to be exposed to these strategies to address vaccine hesitancy without improving access to vaccination services.

Reducing or eliminating costs for vaccines such as the MMR vaccine is one of the most effective ways to improve access and increase vaccine uptake [87]. In the U.S., most children can receive vaccines at no cost through health insurance or the Vaccines for Children (VFC) program, which provides vaccines for individuals who are uninsured, underinsured, have public insurance (Medicaid), or are American Indian/Alaska Native. Since the VFC program was implemented in 1994, MMR vaccination rates in the U.S. have risen significantly to over 90% [1,110]. Globally, the Measles and Rubella Partnership works with countries and populations to ensure sustainable financing for measles and rubella vaccination services [111]. Preventing measles and rubella through vaccination has a very high return on investment. Vaccination scenarios to reach worldwide measles and rubella elimination have been shown to be more cost-effective than current trends for both measles and rubella [112].

Vaccination programs in communities, including schools, childcare centers, community gathering places, and homes, successfully improve vaccine uptake [87]. School- or childcare-based vaccination programs are particularly helpful for children less likely to access healthcare, and this approach has been used in many countries.

### 5.3. Incentives and Requirements to Improve Vaccine Uptake

Incentives and requirements for vaccination are often considered strategies to improve vaccine uptake in those who are willing to be vaccinated. These evidence-based strategies, though, are grounded in behavioral science and often can overcome low to moderate levels of vaccine hesitancy and improve vaccination uptake.

Using incentives has been shown to improve vaccination uptake [87]. For instance, in one RCT, the chance of winning a monetary prize was associated with increased vaccine uptake [113]. Other studies have shown that different incentives, such as tying vaccination to insurance-related incentives or public benefits, can improve vaccine uptake [87]. However, incentive programs may be cost-prohibitive outside of large organizations or insurers. In LMICs, several studies of offering small monetary or non-monetary incentives demonstrated improved vaccine uptake [114,115,116].

Requiring vaccines for schools or childcare attendance can improve vaccine uptake [87]. Many countries have some vaccine requirements for school or childcare attendance with differing degrees of requirements and enforcement [117]. Although there is significant heterogeneity in requirements between countries, several common patterns have been identified. First, the public reaction to vaccine requirements influences the persistence of these requirements within a country. Second, when vaccine-preventable outbreaks occur, this often leads to the introduction of requirements. Third, issues with vaccine access, exemptions, and the enforcement of laws mean the effects of these requirements are variable and context-specific [117].

## 6. Future Directions and Major Research Needs

Although considerable progress has been made in understanding measles and rubella vaccine hesitancy and the evidence for interventions that improve uptake, substantial work remains. Major measles and rubella vaccine hesitancy research needs fall into one of three significant categories of research priorities: (1) further describing the epidemiology and the social and behavioral determinants of vaccine hesitancy, (2) building the evidence for interventions to improve vaccine uptake, and (3) expanding research into LMICs countries and marginalized communities.

Significant work remains to describe the epidemiology of vaccine hesitancy further. We agree with the calls from others to create standardized, timely, and actionable surveillance systems for vaccine hesitancy. These systems need to be grounded in understanding the behavioral and social determinants of vaccine hesitancy, utilize measures validated and used by other countries and jurisdictions, be geospatially granular enough to represent local communities, and be timely and actionable by local vaccination programs [38]. These systems must also be refined in response to evolving evidence, culturally tailored, appropriately representing marginalized communities, and available in multiple languages. Additionally, efforts to address measles and rubella would benefit from a deeper understanding of vaccine hesitancy’s social and behavioral determinants, which inform tailored and culturally relevant interventions.

Communication interventions from healthcare professionals and the implementation of these tools need to be studied further to be improved and optimized. Additionally, evidence for interventions to address the increasing spread of vaccine misinformation, disinformation, and antivaccine activism in online communication is in its infancy, and more research with rigorous methods is needed. Using online communication to improve vaccine confidence also needs to be studied more in-depth to better harness the power of these tools.

Finally, many of the resources described here were studied in well-resourced countries. Research in LMICs is necessary to determine which interventions are the most effective and can be implemented and disseminated sustainably and equitably within these countries.

## 7. Discussion

This article reviewed evidence-based strategies for addressing measles and rubella vaccine hesitancy to improve vaccine uptake and, ultimately, eliminate measles and rubella. The strategies reviewed here should not be viewed as stand-alone nor as immutable. These strategies are best applied in combination and must be tailored to the unique contexts of the communities in which they are being used. Additionally, the evidence for these strategies continues to evolve, as do the recipients of these strategies and the context in which they are delivered. With rapid changes in how information is shared and disseminated, changes in human behaviors and social dynamics, and the lingering effects of the COVID-19 pandemic, ongoing research on the best strategies to address vaccine hesitancy and optimal implementation of these strategies is desperately needed. Vaccine hesitancy is a complex issue that requires cross-discipline collaboration. Funding must be increased for all research investigating attitudinal barriers to vaccination because vaccines alone do not save lives; vaccinations save lives.

Finally, while this article aims to provide background on measles and rubella vaccine hesitancy and an overview of evidence-based strategies for addressing measles and rubella vaccine hesitancy, the problem of access cannot be ignored. The strategies discussed in this article are significantly influenced by the social determinants of health, including equitable access to affordable and quality healthcare services [118]. The availability of health professionals skilled at culturally relevant vaccine communication, health systems with the infrastructure to implement vaccine uptake strategies, health insurance coverage, and reduced or no-cost preventive care services influence the success of vaccine-hesitancy interventions. Tackling these barriers must involve deeper collaboration between healthcare providers, health system leaders, public health professionals, policymakers, and community leaders. Efforts must be made to improve the implementation and availability of evidence-based strategies for addressing vaccine hesitancy to improve vaccine uptake and, ultimately, eradicate measles and rubella worldwide.

## Figures and Tables

**Table 1 vaccines-12-00694-t001:** Healthcare professional communication strategies to address vaccine hesitancy ^1^.

Strategy	Examples/Recommendations
Provide a strong recommendation for vaccination	A strong recommendation from a healthcare professional is one of the best ways to improve vaccine acceptance The strength and quality of the recommendation are important Missed opportunities to recommend vaccination are frequent and correlate with decreased vaccine uptake
Use the presumptive format for initiating vaccine communication	Example: “Today we’re going to do 3 shots” Example: “John’s due for 2 vaccines today” Example: “I know you have had concerns before, but Sara is due for 3 shots today” Tone and body language matter A presumptive format can be used even if resistance has been voiced previously
Pursue adherence despite initial reluctance	Example: “The MMR vaccine is very important for Jack to receive” Pursuing adherence is in contrast to immediately acquiescing when parents express initial reluctance
Use motivational interviewing for individuals who express reluctance	Open ended questions: try to understand the individual’s stance on vaccination. Example: “Can you tell me more about what you’ve heard?” Affirmations: help the individual to feel supported, appreciated, and understood. Example: “It is clear that you care about your child’s health.” Reflections: reflect the individual’s words to encourage partnership and build rapport. Example: “It sounds like you are worried about side effects of the MMR vaccine.” Ask Permission to Share: ask before sharing information to increase individual’s receptiveness. Example: “Can I share what I know about the MMR vaccine with you?” Autonomy Support: enhance the individual’s sense of control. Example: “Ultimately, this is a decision only you can make.”

^1^ Adapted from: O’Leary ST, Opel DJ, Cataldi JR, et al. Strategies for Improving Vaccine Communication and Uptake. Pediatrics; 2024 [27].

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
