# Peer review of "A World without Measles and Rubella: Addressing the Challenge of Vaccine Hesitancy"

_vaccines, 2024, doi:10.3390/vaccines12060694_

Round 1

Reviewer 1 Report

Comments and Suggestions for Authors

The manuscript entitled “A World without Measles and Rubella: Addressing the Challenge of Vaccine Hesitancy” by Higgins and O’Leary submitted to the journal Vaccines (manuscript ID 2997581) has been reviewed. The manuscript provides a review on vaccine hesitancy and strategies addressing hesitancy and future directions and major areas of research. It is well written, and the review appears solid. A few minor comments from the reviewer mainly pertaining to the length of the manuscript which can be condensed significantly in some of the paragraphs. Table 1 is already to some degree mentioned in the text and can be removed. 

Line 253 onwards and 299 onwards: overall, the review is based on studies predominantly conducted in high income countries, mainly North America. However, vaccine hesitancy has become an issue in many countries and authors should try to address not only HIC but also LMIC. Understanding the limited literature that may be available on LMIC. 

Line 356 on: paragraph 4.1 can be condensed significantly

Line 465 on: paragraph 4.3 on mass communication should expand on the role of social media as it seems a major contributor to the spread of vaccine hesitancy, not only in the West but also in LMIC. 

Line 529: par 5.2 reviewer is uncertain if this paragraph needs to be included. 

Line 553: par 5.3 same here. Authors to provide more arguments whether to include or not. 

Reviewer 2 Report

Comments and Suggestions for Authors

The authors have submitted a reasonable review for a planned special issue. As a review, the manuscript does not contain new data but does organize useful insights. There are a few ways in which the manuscript could be strengthened, however.

The authors make a case for the use of various strategies in addressing vaccine hesitancy in the case of measles and rubella. What they have not done sufficiently to contextualize the specific cases of measles and rubella within the larger landscape of vaccination literature. To what extent, if at all, do the disease cases pose unique challenges? What other disease cases are most similar to measles and rubella specifically? The authors should more directly articulate comparison with other types of vaccination to optimize the utility of this paper.

The authors begin to address possibilities for strategies to encourage healthcare professional behavior and training for healthcare professionals. There are two dimensions of that literature which warrant more attention, however.

First, the authors concentrate mostly on first-time encounters between professionals and patients regarding vaccination. They do acknowledge the importance of persistence but do not extensively discuss the literature on follow-up and long-term relationships. It would be useful to cite some of the emerging literature on that topic, e.g., Margolis, M. A., Brewer, N. T., Boynton, M. H., Lafata, J. E., Southwell, B. G., & Gilkey, M. B. (2022). Provider response and follow-up to parental declination of HPV vaccination. Vaccine, 40, 344-350.

Second, the authors outline the potential importance of tactics such as motivational interviewing but they could further discuss available literature on the need for training healthcare professionals to use those approaches, e.g., Wood, J., Lee, G., Stinnett, S., & Southwell, B. (2021). A pilot study of medical misinformation perceptions and training among practitioners in North Carolina (USA). INQUIRY: The Journal of Health Care Organization, Provision, and Financing, 58, 469580211035742. doi: 10.1177/00469580211035742. 

Making these revisions would strengthen the value of the manuscript as a review of relevant literature in important ways. 

Round 2

Reviewer 2 Report

Comments and Suggestions for Authors

The authors have addressed reviewer comments. One of the citations appears to be a duplicate but beyond that the manuscript is improved. 

Author Response

Thank you for your observation of the duplicated reference. We have corrected this (references #40 and #48 were duplicated in the citation manager).